# A somatic piRNA pathway in the *Drosophila* fat body ensures metabolic homeostasis and normal lifespan

Brian C. Jones[1], Jason G. Wood[1], Chengyi Chang[1], Austin D. Tam[1], Michael J. Franklin[1], Emily R. Siegel[1] & Stephen L. Helfand[1]

In gonadal tissues, the Piwi-interacting (piRNA) pathway preserves genomic integrity by employing 23–29 nucleotide (nt) small RNAs complexed with argonaute proteins to suppress parasitic mobile sequences of DNA called transposable elements (TEs). Although recent evidence suggests that the piRNA pathway may be present in select somatic cells outside the gonads, the role of a non-gonadal somatic piRNA pathway is not well characterized. Here we report a functional somatic piRNA pathway in the adult *Drosophila* fat body including the presence of the piRNA effector protein Piwi and canonical 23–29 nt long TE-mapping piRNAs. The *piwi* mutants exhibit depletion of fat body piRNAs, increased TE mobilization, increased levels of DNA damage and reduced lipid stores. These mutants are starvation sensitive, immunologically compromised and short-lived, all phenotypes associated with compromised fat body function. These findings demonstrate the presence of a functional non-gonadal somatic piRNA pathway in the adult fat body that affects normal metabolism and overall organismal health.

[1] Department of Molecular Biology, Cell Biology, and Biochemistry, Brown University, Providence, Rhode Island 02912, USA. Correspondence and requests for materials should be addressed to S.L.H. (email: Stephen_Helfand@Brown.edu).

Transposable elements (TEs) parasitize the DNA of their hosts and account for a large portion of eukaryotic genomes[1,2]. To combat the invasion and expansion of TEs, small RNA (smRNA) silencing pathways have evolved to suppress TEs across species from plants to humans[3]. The short interfering RNA pathway suppresses TEs in all tissues of plants and animals, whereas the activity of the Piwi-interacting RNA (piRNA) pathway is thought to be primarily restricted to the gonads of metazoans[4,5]. Loss or decline of these pathways results in genomic instability and cellular dysfunction caused by TE reactivation and transposition[6–9].

The piRNA pathway is best known for its role in gonadal tissues where it protects against genomic damage caused by TE reactivation[4,5]. The pathway silences TEs by employing complementary small RNAs called piRNAs, generated from large TE-rich genomic regions called piRNA clusters. In flies, these clusters transcribe long single-stranded RNA precursors that are then further processed into smaller 23–29 nucleotide (nt) piRNAs. These piRNAs partner with argonaute effector proteins (Piwi, Aubergine or AGO3) that are then able to silence TEs via their homology to TE transcripts[4,10]. This process is accomplished by one of two silencing mechanisms. In the primary piRNA pathway, active in both the germline and ovarian somatic follicle cells, Piwi represses TE transcription by establishing heterochromatin[5,11]. In the secondary piRNA pathway, active only in the germline, Aubergine and AGO3 silence TEs post transcriptionally in the cytoplasm via messenger RNA cleavage[4,5,10]. Although the role of the piRNA pathway was previously thought to be restricted to the gonads, recent evidence in a diversity of organisms suggests that this pathway may also be present in somatic cells outside of the gonad[12].

Over the past decade, new evidence has begun to reveal non-gonadal examples of the piRNA pathway including a role for the piRNA pathway in stem cell function[12]. In planaria, piRNA pathway proteins are essential in maintaining stem cell pluripotency as well as the regenerative capacity of these animals[13]. piRNA pathway components are active in multiple types of cancer[12], including specific cancers in mammals and flies[12,14–18], and in flies piwi has been shown to contribute to malignant tumour growth[19]. Less information is available for a role of the piRNA pathway in normal differentiated somatic tissues, although evidence for the activity of the secondary piRNA pathway in specific neurons of the adult fly brain has been reported[20]. As more non-gonadal examples of an active RNA interference system are discovered, it appears that the piRNA pathway may have other important roles beyond its known functions in gonadal tissue.

Here we show the presence of a functional somatic piRNA pathway in the adult fly fat body. The piRNA pathway in the fat body exhibits all the canonical characteristics of a primary piRNA pathway and actively suppresses TE mobilization in this tissue. We observe that loss of this pathway correlates with compromised fat body function and shortened lifespan. These findings demonstrate a novel role for the piRNA pathway outside of the gonads in a fully differentiated somatic tissue.

## Results

**The fat body exhibits components of an intact piRNA pathway.** Given the recent evidence that the piRNA pathway may also be active in select non-gonadal somatic cells[12], we examined the expression of piRNA pathway genes outside of the gonads. We found that examination of RNA sequencing (RNA-seq) libraries from adult fly eviscerated abdomen, but not heads or thorax, had significant enrichment of piRNA pathway genes relative to other somatic body segments (Fig. 1a, Supplementary Fig. 1, and

Supplementary Dataset 1). Immunofluorescent microscopy demonstrated specific localization of the Piwi protein to the nuclei of adult abdominal and pericerebral fat body cells, but not to nuclei of cells from other tissues outside of the gonads or in fat bodies of homozygous piwi null mutants (Fig. 1b). Immunoblotting of isolated purified fat body, thorax and head also demonstrated the presence of Piwi protein in the fat body (Fig. 1c and Supplementary Fig. 2). The presence of Piwi protein in the fat body suggests the possibility of an intact piRNA pathway in this tissue.

Activated piRNAs in functional piRNA silencing complexes are 2′-O-methylated at their 3′ termini and can be selectively enriched and detected using periodate oxidation[4,21]. We found a broad piRNA-like peak of smRNAs ranging from 23 to 29 nt in oxidized small RNA-seq (smRNA-seq) libraries from pure adult abdominal fat body (Fig. 1d), suggesting the presence of piRNAs with 2′-O-methylation at their 3′ termini as is typical of gonadal piRNAs actively loaded into a piRNA-argonaute complex[22]. Of the oxidized fat body piRNAs, 49% (23–29 nt smRNAs) mapped to TEs (Fig. 1e) and these reads exhibited a strong antisense bias (Supplementary Fig. 3a) and a canonical first position nucleotide bias for uracil[10] (Fig. 1f), indicating that they likely have the capacity to target TE transcripts for silencing. Oxidized smRNA libraries from purified abdominal fat bodies of two distantly related drosophilids, Drosophila simulans and Drosophila yakuba (Supplementary Fig. 3b), also showed 23–29 nt piRNAs mapping to TEs (Supplementary Fig. 3c–f) with antisense TE-mapping piRNAs exhibiting a strong first position uracil bias (Supplementary Fig. 3g,h), demonstrating that adult fat body piRNAs are conserved across diverse drosophilid species. Together, these data suggest the presence of fat body piRNAs that exhibit canonical piRNA characteristics, are associated with an active piRNA-argonaute silencing complex and are evolutionarily conserved.

**The fat body piRNA pathway suppresses TEs.** We next examined whether the fat body piRNA pathway exhibits canonical hallmarks of an active piRNA pathway. Loss of Piwi in gonadal tissues results in a dramatic reduction of piRNAs and a derepression of their corresponding TEs[11,23]. In the fat bodies of piwi null mutants, we observed a significant increase in the transcript levels of multiple TEs and a corresponding decrease of their associated piRNAs (Fig. 2a and Supplementary Fig. 4). Total piRNAs also decreased (28.1% decrease) as well as TE-mapping piRNAs (70.5% decrease) (Fig. 2b, side panel). We used a reporter of transposition for the endogenous gypsy retrotransposon (gypsy-TRAP)[9,24], a known target of Piwi in the ovary[25], to detect transposition in fat body cells and found that piwi mutant fat bodies displayed significantly higher levels of gypsy transposition relative to controls (Fig. 2c). Together, these data show that a somatic piRNA pathway actively suppresses the expression and mobilization of TEs in adult fat body cells.

The piRNAs that target and suppress TEs in ovarian tissues are known to originate from genomic regions called piRNA clusters[10]. Mapping fat body piRNAs to previously annotated fly piRNA clusters showed that many map to the flamenco cluster, which is known to be specific to somatic follicle cells (Fig. 2d,e)[10,11]. In agreement with studies of ovarian follicle cells[10,11], piRNAs mapping to flamenco were depleted in the fat bodies of piwi mutants (Fig. 2d,e). These data further support the presence of a functional fat body piRNA pathway where piRNAs produced from somatic piRNA clusters pair with Piwi to suppress TEs as the primary piRNA pathway does in ovarian follicle cells[10,11].

Previous studies in flies have shown that primary piRNAs are also derived from the 3′ untranslated regions (UTRs) of coding genes[26,27]. We observed that 17% of fat body piRNAs from the

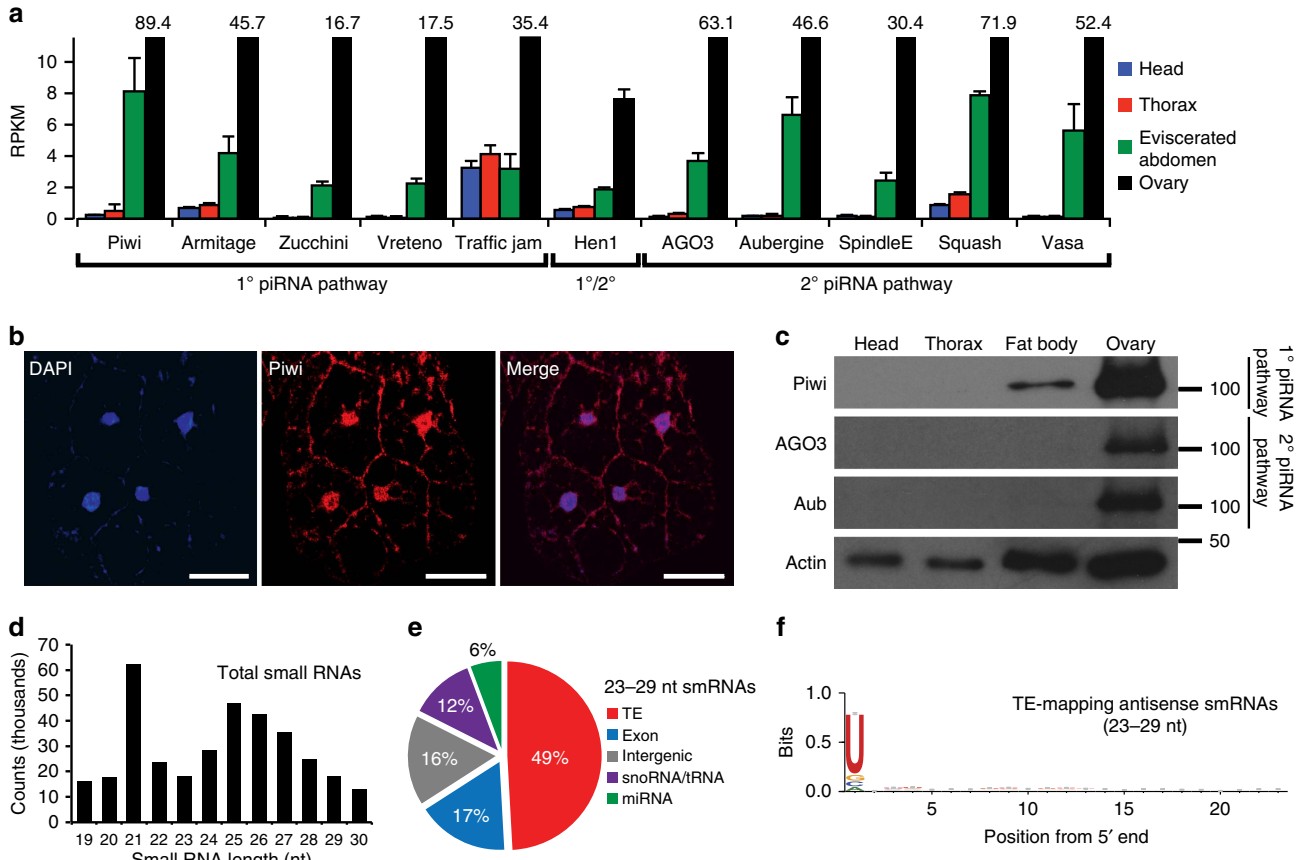

**Figure 1 | Canonical signatures of a piRNA pathway in the adult fly fat body. (a)** Expression of primary and secondary piRNA pathway genes generated from total RNA-seq libraries of head, thorax, eviscerated abdomen and ovary. piRNA pathway genes are more highly expressed in the eviscerated abdomen than in the head or thorax. Data values for ovary libraries that exceed the range of the plot are shown above each relevant bar. RPKM, reads per kilobase per million. Error bars represent s.e.m.; $n = 3$ replicate libraries. In comparing the eviscerated abdomen with head and thorax controls, 10 of 11 genes (excluding *tj*) are statistically significant ($P < 0.0001$). See Supplementary Dataset 1 for statistics. **(b)** Piwi protein localizes to the nuclei of abdominal fat body cells. DAPI labels fat body nuclei. Staining in the membrane is autofluorescence typical of fat body cells. Scale bars represent 20 μm. **(c)** Piwi protein is present in the fat body. All piRNA argonautes are present in the ovary samples. Actin serves as a loading control. **(d)** Fat body smRNA size profile from oxidized smRNA-seq libraries. Oxidation allows for enrichment of 2′-O-methylated smRNAs. Peak at 21 nt likely represents short interfering RNA (siRNA) population. Broader peak from 23 to 29 nt represents putative fat body piRNAs. Reads aligning to rRNA and miRNA were excluded from analysis. **(e)** Fat body piRNAs (23–29 nt) aligned to the fly genome map primarily to TEs. **(f)** Sequence composition of TE-mapping fat body piRNAs (23–29 nt) displays a first position nucleotide bias for uracil.

oxidized fat body library mapped to coding genes (Fig. 1e) and that 3′UTR sense-mapping piRNAs were depleted in *piwi* mutants (17.5% decrease; Fig. 2b, side panel). These 3′UTR-derived piRNAs are between 78% ($piwi^{-/+}$) and 87% (wt) sense-mapping, as is expected of genic piRNAs (Supplementary Fig. 5a). Fat body piRNAs mapping to the 3′UTR of *traffic jam* (*tj*), a known source of genic piRNAs in ovarian follicle cells[26,27], were depleted in *piwi* mutants (Fig. 2f,g and Supplementary Dataset 2) along with other 3′UTR-derived piRNAs (Supplementary Fig. 5b–e and Supplementary Dataset 2). Together, these data provide evidence of 3′UTR-derived fat body genic piRNAs, another hallmark of the primary piRNA pathway.

In order to rule out the possibility that fat body piRNAs result from contamination by ovarian tissues during dissection and library preparation, we performed smRNA-seq on oxidized fat body libraries isolated from $ovo^{D1}$ flies. Because of a dominant female-sterile mutation in the *ovo* gene, these flies exhibit severely degenerated ovaries, thus significantly decreasing the likelihood that any piRNAs isolated from the fat body of these animals would be due to contamination from ovarian tissues. We observed in both wild-type and $ovo^{D1}$ fat bodies 23–29 nt

smRNAs that mapped uniquely to the *flamenco* locus (Supplementary Fig. 6a,b), suggesting that piRNAs observed in oxidized fat body libraries are in fact originating from this tissue and not a result of ovarian contamination. In addition, immunoblotting demonstrates the presence of Piwi protein in the eviscerated abdomen and isolated fat body of $ovo^{D1}$ flies (Supplementary Fig. 6c). These data, combined with our observation of the Piwi protein in the nuclei of fat body cells (Fig. 1b and Supplementary Fig. 2), strongly support the presence of a functional canonical piRNA pathway in the fat body.

**DNA damage and metabolic dysregulation in piRNA mutants.** The replicative mobility of TEs can contribute to mutagenesis via their ability to insert into new genomic loci[6]. TE reactivation and transposition has been shown to correlate with chromosomal rearrangements, double-strand DNA breaks and apoptosis[7,8]. Phosphorylation of the histone variant H2A.v (γ-H2A.v) during DNA repair serves as a reliable marker of double-strand DNA breaks and has been shown to correlate with increased TE activity in the fat body[9,28].

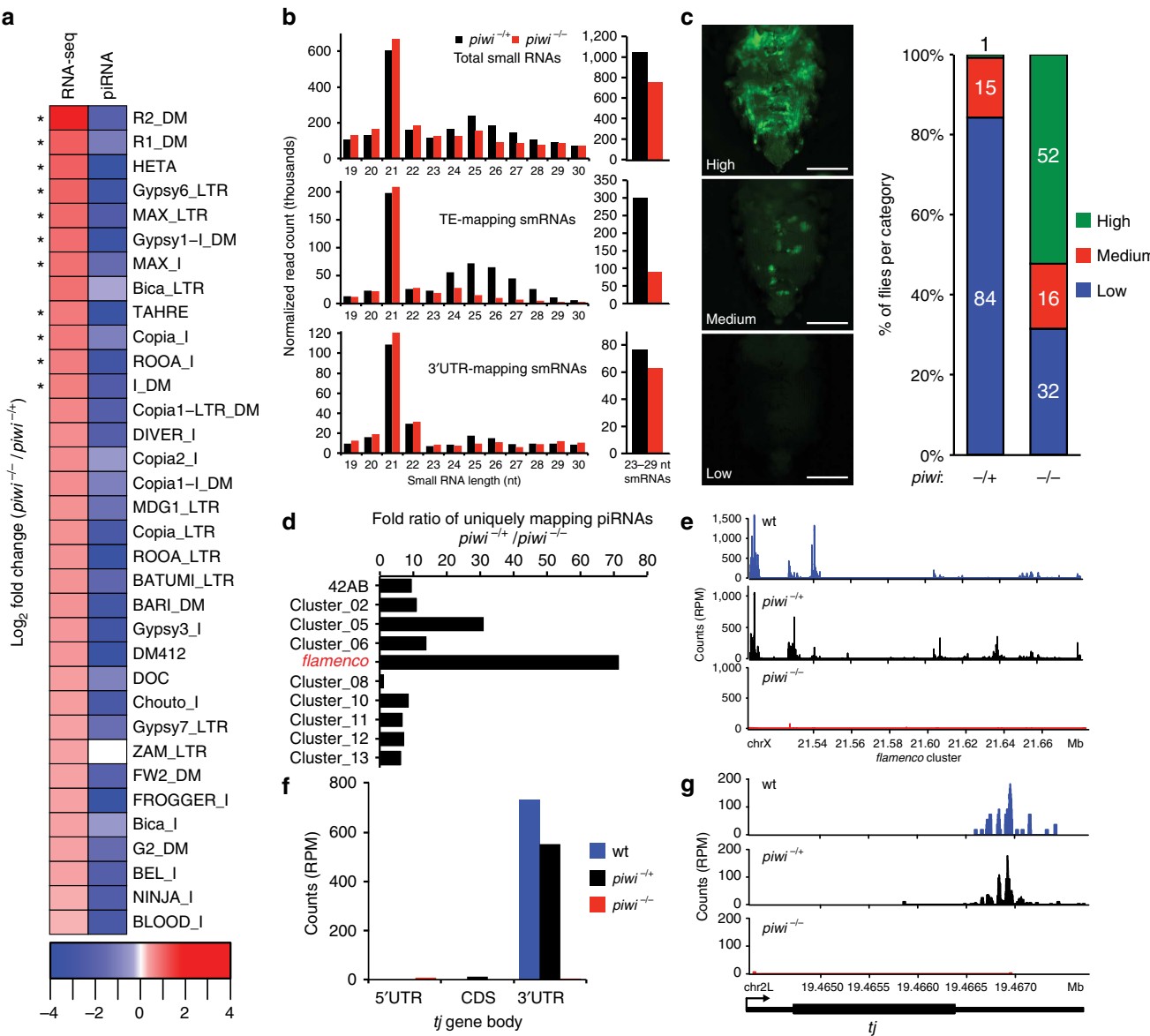

**Figure 2 | The piRNA pathway is active in the fat body.** (**a**) Heat map of log$_2$ fold change of TE transcript levels (total RNA-seq) >1.2 fold change and corresponding piRNAs (smRNA-seq) in *piwi* mutant (*piwi$^{-/-}$*) compared with heterozygous control (*piwi$^{-/+}$*) fat bodies; *n* = 3 replicate libraries. *False discovery rate (FDR) <0.05. (**b**) Fat body smRNA size profile from oxidized smRNA-seq libraries of *piwi* mutant fat bodies (red) and heterozygous controls (black). Shown are total smRNAs (top), TE-mapping smRNAs (middle) and 3'UTR-mapping smRNAs (bottom). Panels at right show levels for 23–29 nt smRNAs for each genotype. (**c**) A fat body-specific transposition reporter line, *gypsy-TRAP/r4-GAL4::UAS-GFP*, in a *piwi* mutant background (see Methods). GFP-positive cells are cells in which a transposition event has activated reporter function. The 10-day-old *piwi* mutants show elevated levels of GFP-positive cells compared with heterozygous controls. Images are of representative flies exhibiting high, medium and low levels of GFP-positive cells (left panels). Scale bars represent 150 µm. Panel on right shows the distribution of flies for each group by genotype; numbers within each bar are corresponding percentages for each group. Fisher's exact test compared the combined 'High' + 'Medium' groups and the 'Low' group of each genotype. *piwi$^{-/+}$*: *n* = 140 flies; *piwi$^{-/-}$*: *n* = 142 flies; *P* < 0.0001. (**d**) Fold ratio of uniquely mapping cluster piRNAs in *piwi* mutants compared with heterozygous controls. The *flamenco* cluster shows the greatest response to loss of *piwi*. (**e**) Unique piRNA reads (23–29 nt) map to the *flamenco* locus in wild-type (wt) and *piwi* heterozygotes and are lost in *piwi* mutants. (**f,g**) Unique piRNA reads (23–29 nt) map to the *tj* gene body in wt and *piwi* heterozygotes and are lost in *piwi* mutants. Thick lines in gene model (**g**) represent coding sequence, and thin lines represent 5' and 3'UTRs. See Supplementary Dataset 2 for raw data.

Using immunofluorescent microscopy, we observed an increase in the intensity of γ-H2A.v staining in *piwi* mutants relative to controls (Fig. 3a,b). These data suggest that the fat body piRNA pathway normally protects fat body cells from the accumulation of DNA damage that may be caused by TE reactivation.

The fly fat body is a functional analogue of the mammalian liver and adipose tissue, with one of its primary roles being storage of lipids and glycogen[29]. Our observation that *piwi* mutant fat body cells showed increased TE mobilization and elevated levels of DNA damage (Figs 2c and 3a,b) led us to hypothesize that this could result in disrupted fat body function. Nile Red staining of fat body lipid droplets revealed a reduction of lipid droplet size in *piwi* mutants compared with controls, with larger lipid droplets (>200 µm$^2$) greatly reduced in their abundance (Fig. 3c–e and Supplementary Dataset 3). These data correlate with a significant reduction of two of the major

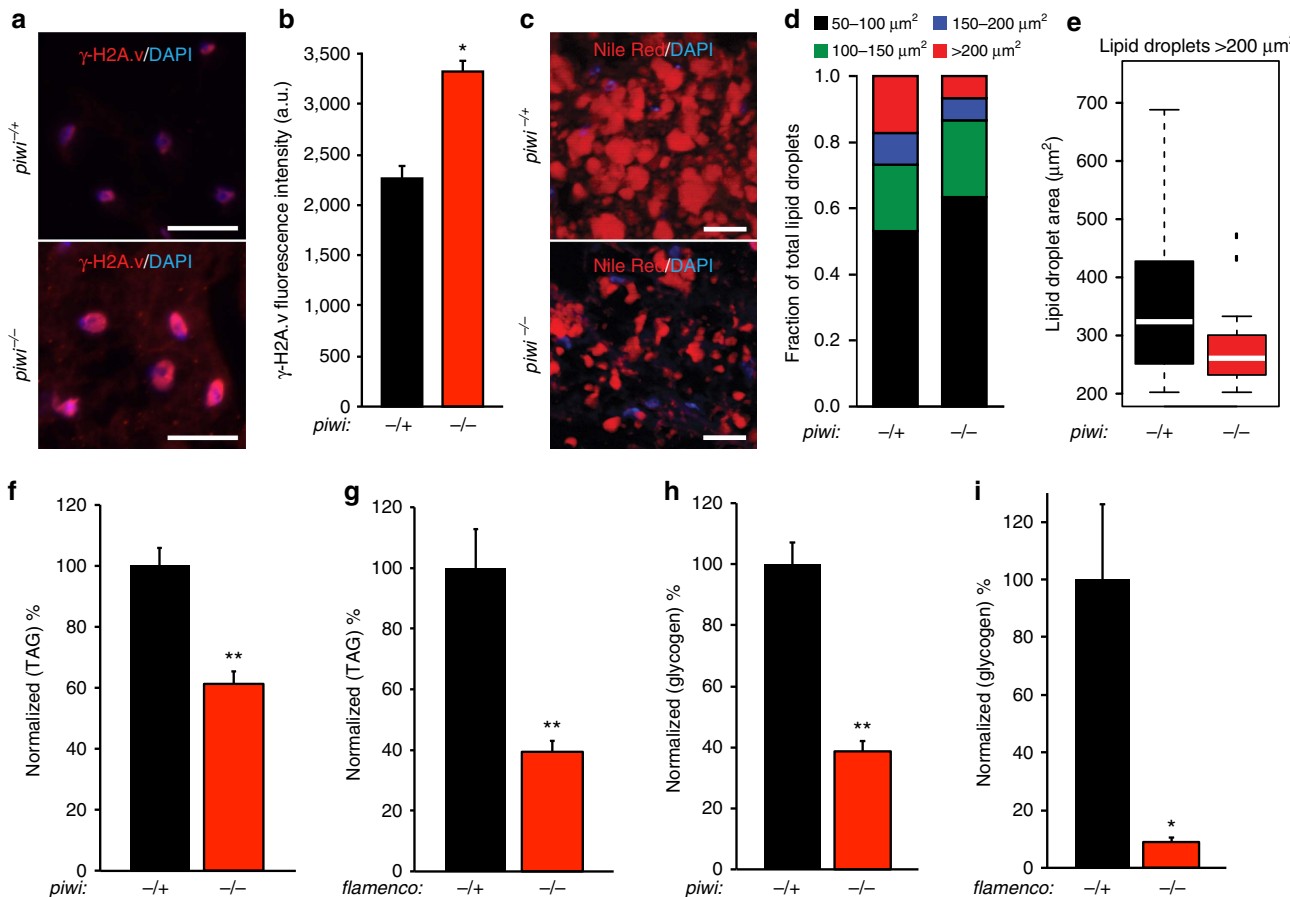

**Figure 3 | Loss of the piRNA pathway disrupts normal metabolic functions of the fat body.** (**a**) Representative images of γ-H2A.v staining in *piwi* mutants and heterozygous controls. *piwi* mutants exhibit higher levels of γ-H2A.v staining compared with heterozygous controls. Scale bars represent 20 μm. (**b**) Quantification of γ-H2A.v staining in (**a**). *piwi*$^{-/+}$ and *piwi*$^{-/-}$: error bars are s.e.m. Student's two-tailed *t*-test compared with heterozygous control; *n* = 104 nuclei per genotype. *$P$ < 0.0001. (**c**) Representative images of Nile red staining of fat body lipid droplets in *piwi* mutants and heterozygous controls. *piwi* mutants exhibit smaller lipid droplets relative to heterozygous controls. Scale bars represent 20 μm. (**d**) Quantification of lipid droplet staining in (**c**). *piwi*$^{-/+}$: *n* = 6 flies, *piwi*$^{-/-}$: *n* = 5 flies. See Supplementary Dataset 3 for raw data. (**e**) Box plot showing distribution of lipid droplets > 200 μm$^2$ from (**d**) comparing *piwi* mutants to heterozygous controls. (**f**–**i**) Measurements of whole-body adult fly TAGs (**f**,**g**) and glycogen (**h**,**i**) of *piwi* (**f**,**h**) or *flamenco* (**g**,**i**) mutants compared with heterozygous controls. Data were normalized to total protein concentration of each sample and represented as a percent of the heterozygous control. Error bars are s.e.m. For each assay, *n* = 5 biological replicates per genotype. Student's two-tailed *t*-test compared with heterozygous control. *$P$ < 0.01; **$P$ < 0.001. All assays were performed using 10-day-old flies.

storage metabolites in fat body, triacylglycerides (TAGs) and glycogen, in both *piwi* and *flamenco* mutants (Fig. 3f–i and Supplementary Fig. 7a,b). We next asked whether these phenotypes correlated with an altered fat body transcriptome as transcription of TEs alone can cause RNA toxicity and contribute to cellular dysfunction[8]. We generated RNA-seq libraries from *piwi* mutant and control fat bodies and found that many differentially expressed genes in metabolism-associated pathways were significantly changed (Supplementary Fig. 7c). These data support a model in which a loss of the piRNA pathway function results in a decrease in lipids and stored metabolites, disruption of metabolic homeostasis and a decline in cellular function, possibly because of reactivation of TEs.

**piRNA mutants show altered fat body function and lifespan.** The fly fat body plays a major role in resisting stressors such as fasting and, through its role in innate immunity, in resisting pathogenic infections[29]. We observed that *piwi* mutants were highly sensitive to starvation conditions as well as infection by a pathogenic insect bacterium (Fig. 4a,b and Supplementary

Table 1). Because of the central role that the fat body plays in regulating longevity[30,31], we next examined the effect of disrupting the piRNA pathway on lifespan. We found that mutations in either *piwi* or the *flamenco* locus dramatically shorten lifespan (Fig. 4c,d and Supplementary Table 1). Finally, we tested whether the shortened lifespan of piRNA pathway mutants was dependent upon TE activity. Many TEs in *Drosophila* are retrotransposons, including *gypsy*, and depend upon reverse transcriptase for their replicative mobility[32]. Administration of a known reverse transcriptase inhibitor, 3TC, inhibits the normal age-related increase in *gypsy* mobilization and extends the shortened lifespan of *Dcr-2* mutants, another condition in which derepression of TEs occurs[9]. We administered 3TC to *flamenco* mutants and observed a significant lifespan extension (Fig. 4e and Supplementary Table 1), suggesting that a shortened lifespan phenotype is at least partially dependent upon TE mobilization. These results suggest that loss of the fat body piRNA pathway and an increase in TE activity and mobilization correlates with compromised fat body function including its ability to otherwise mitigate the detrimental effects of environmental stressors.

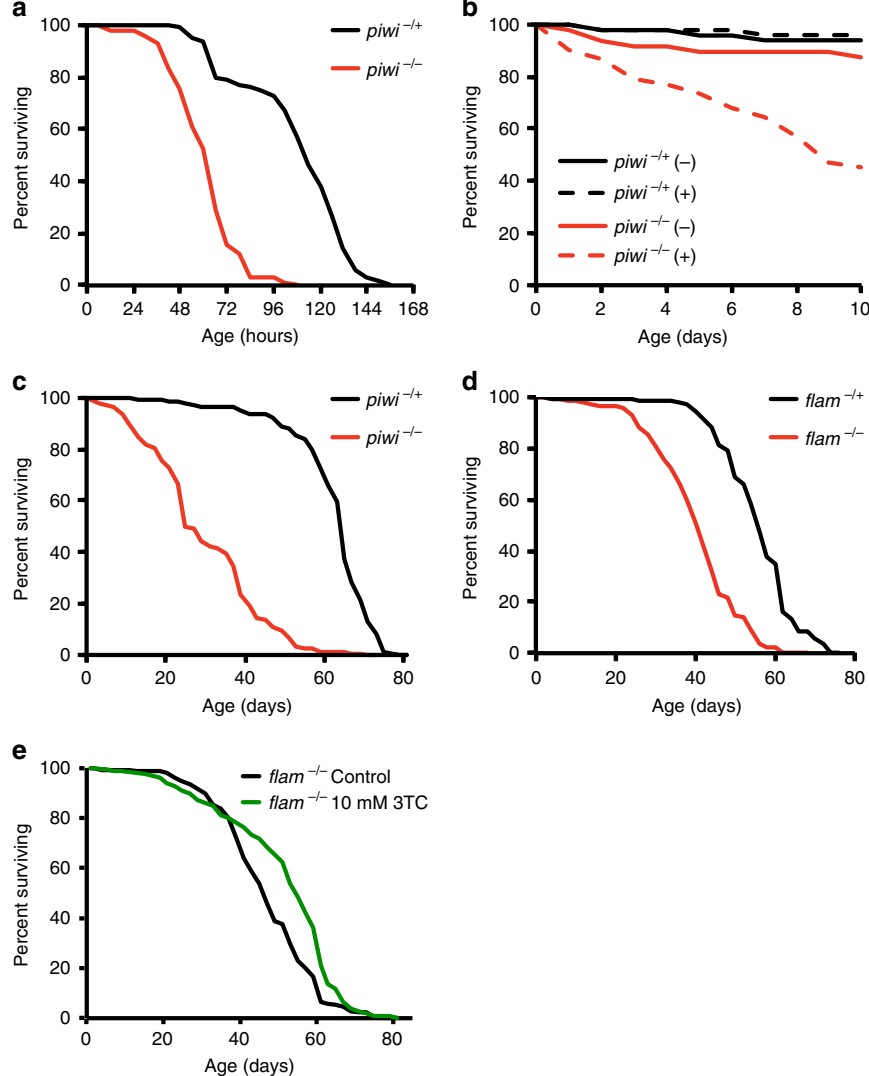

**Figure 4 | piRNA pathway mutants are stress sensitive and short-lived.** (**a**) Survivorship curves for starvation of *piwi* mutants and heterozygous controls. *piwi* mutants are more sensitive to starvation than heterozygous controls. Log rank test compared with heterozygous controls; $n \approx 50$; $P < 0.0005$. (**b**) Survivorship curves for immune challenge of *piwi* mutants and heterozygous controls. *piwi* mutants are more sensitive to infection than heterozygous controls. Flies were either infected with a mock EtOH control ( − ) or a culture of *E. carotovora* ( + ). Log rank test compared with heterozygous or mock EtOH control ( − ); $n \approx 50$; $P < 0.0005$. (**c**) Survivorship curves of *piwi* mutants and heterozygous control. *piwi* mutants are shorter lived compared with heterozygous controls. Wilcoxon rank sum test compared with heterozygous control; $n \approx 250$; $P < 0.0005$. (**d**) Survivorship curves of *flam* mutants and heterozygous control. *flam* mutants are shorter lived compared with heterozygous controls. Wilcoxon rank sum test compared with heterozygous control; $n \approx 250$; $P < 0.0005$. (**e**) Survivorship curves of *flam* mutants fed 10 mM 3TC. *flam* mutant flies fed 3TC live longer than untreated controls. Wilcoxon rank sum test compared with control; $n \approx 250$; $P < 0.0005$. See Supplementary Table 1 for assay parameters and statistics. All assays were repeated twice with similar results.

## Discussion

Here we have shown evidence for a fully functional piRNA pathway in a non-gonadal somatic tissue, the adult fly fat body, that is likely to be necessary for proper tissue function and overall organismal health. These results demonstrate that the adult fat body piRNA pathway exhibits canonical characteristics found in gonadal somatic cells, and its activity likely positively affects the function of a tissue important to metabolic homeostasis and physiological health. Although we are not able to entirely rule out a contribution of the gonadal piRNA pathway to fat body function, many of the phenotypes we observe are opposite to those typically seen in animals with compromised gonadal tissue function and therefore likely represent the effect of a loss of the fat body piRNA pathway. For example, the shortened lifespan and reduced lipid stores in piRNA pathway mutants

demonstrates that the piRNA pathway is essential in the health and functioning of non-gonadal somatic tissues, as reduction or ablation of gonadal function in flies often extends lifespan and increases lipid stores rather than decreasing lifespan and fat storage[31]. Recent studies in wild-type flies have also demonstrated an important link between TE activity and longevity[9], and our studies demonstrating partial rescue of the shortened lifespan in *flamenco* mutants upon administration of a reverse transcription inhibitor further support this association.

Interest in a function for the piRNA pathway in the soma has increased recently as new roles for this pathway are being illuminated. The piRNA pathway's association with tissues that maintain a degree of immortalization similar to that exhibited in the germline is of particular interest[12]. For example, the somatic stem cell niches of Hydra maintain an active piRNA pathway that

represses TEs, possibly contributing to this organism's remarkably long lifespan[33]. These studies, together with our findings, suggest that the presence of a piRNA pathway in normal somatic tissues may offer an additional cellular defence against TE reactivation and possible somatic genomic damage. Our finding of a role for the piRNA pathway in preserving metabolic homeostasis and the overall health of the fly suggests the potential importance of the piRNA pathway in other somatic tissues. Finally, interventions specifically augmenting the piRNA pathway may provide significant benefits to maintaining genomic integrity, tissue function and healthy lifespan.

## Methods

**Fly stocks and husbandry.** *Drosophila* stocks were all maintained on standard media (30.5 g l$^{-1}$ autolysed yeast, 121.8 g l$^{-1}$ sucrose, 52.3 g l$^{-1}$ cornmeal, 8.75 g l$^{-1}$ agar and 2.62 g l$^{-1}$ tegosept, all weight by volume) at 25 °C with a 12-h light/dark cycle at 60% relative humidity. Unless otherwise noted, flies used in all experiments were collected over a 48-h period, placed in density-controlled, mixed-sex vials and aged for 10 days on food containing 150 g l$^{-1}$ autolysed yeast, 150 g l$^{-1}$ sucrose and 20 g l$^{-1}$ agar, all weight by volume. Unless otherwise noted, all experiments were performed using mated female flies grown under these conditions.

Lab stocks of Canton S or $w^{1118}$ were used for wild type experiments. $flam^{KG}/FM4$ (Bloomington 16453) was obtained from the Bloomington *Drosophila* Stock Center at Indiana University. Stocks of *D. yakuba* and *D. simulans* were provided by the *Drosophila* Stock Center at the University of California, San Diego. The *piwi* mutant fat body *gypsy-TRAP* reporter line was generated from the *gypsy-TRAP* line provided to us by Joshua Dubnau[24], *r4-GAL4* (Bloomington 33832), *UAS-GFP* (Bloomington 1522) and the *piwi²/CyO* line from Haifan Lin. $ovo^{D1}$ mutants lacking ovaries were generated by crossing males of $ovo^{D1}$ ($ovo^{D1}v^{24}$/C(1)DX, $y^1w^1f^1$) (Bloomington 1309) to virgin Canton S females.

**RNA and smRNA-seq library preparations.** Flies were flash frozen on dry ice and head and thorax body segments were dissected on a $-20$ °C cold block and stored at $-80$ °C. Eviscerated abdomens and pure adult fat bodies were dissected from the abdominal wall in cold phosphate-buffered saline (PBS) and also stored at $-80$ °C.

Total RNA was extracted from relevant fly tissues using the mirVana miRNA Isolation Kit (ThermoFisher Scientific AM1560). For RNA-seq library prep, 100 ng of total RNA was used as input for the Ovation Universal RNA-seq System kit (Nugen 0343), with *Drosophila* ribosomal RNA (rRNA) depletion module, according to the manufacturer's instructions. Three biologically independent libraries were made for each tissue. For smRNA libraries, 2 μg of total RNA was oxidized in 25 mM sodium periodate with 30 mM borax and 30 mM boric acid (pH 8.6) for 30 min at room temperature (Phillip Zamore Lab Illumina TruSeq Small RNA Cloning Protocol April, 2014; http://www.umassmed.edu/zamore/resources/protocols/). RNA was then recovered with the RNA Clean & Concentrator-5 kit (Zymo Research R1015) and used as input to the NEBNext Multiplex Small RNA Library Prep Set for Illumina (New England Biolabs E7300), with modifications adapted from[34] (2S rRNA block oligo added before 5′ ligation step to decrease rRNA reads in final library).

**Tissue RNA-seq.** Reads were mapped to the dm3 genome using Tophat. The Bioconductor R package easyRNASeq was used to count reads (using 'geneModels' summarization parameter) and calculate normalized reads per kilobase per million (RPKM) values. Error bars presented represent s.e.m. with three independent biological replicates for each condition. The false discovery rate values in Supplementary Dataset 1 were calculated using the Bioconductor edgeR package, with a GLM correcting for batch effects as described in the vignette.

**smRNA size profiles and nucleotide bias.** Oxidized small RNA libraries were first trimmed of adapter sequences using cutadapt and then aligned against *Drosophila* rRNA sequences using Bowtie to remove rRNA aligning reads. For pie charts, reads were first aligned to the dm3 genome, keeping only reads that aligned. Reads were next consecutively mapped to the following genomic compartments using Bowtie ($-v$ 1): sno + tRNA, microRNA (miRNA), TEs, exons and intergenic regions (all sequences downloaded from FlyBase, except TEs that came from Repbase). Reads aligning to each compartment were counted to create a chart.

For total small RNA size profiles (see, for example, Fig. 1d), miRNAs were first removed using Bowtie ($-v$ 1), and remaining reads were subsequently fed into a Perl script that counted the number of reads for each different length between 18 and 50 nucleotides. For size profiles of TE-mapping reads (Supplementary Fig. 2), reads were aligned to the set of Repbase *Drosophila* consensus sequences using Bowtie ($-v$ 1 $-k$ 1--best), and sorted into sense/antisense (with respect to TE) using samtools. Alignments were then converted back into FASTQ files using bedtools, and size profiles calculated as described above. For *D. simulans* and

*D. yakuba* pie charts and size profiles, analysis was performed as described, using relevant genomic compartments for each species downloaded from Flybase. For transposable elements, the set of all *Drosophila* species transposons from Repbase was used.

For nucleotide bias calculations, 23–29 bp TE-mapping antisense alignments were converted into FASTA format using bedtools and EMBOSS, trimmed to uniform length of 23 bp using FASTX-toolkit and used as input for the WebLogo 3 program[35].

**TE and coding gene analysis.** In order to properly account for multi-mapping RNA-seq reads when analyzing TEs, we used the RepEnrich approach[36] to quantify read counts for each TE. This approach combines all instances of each annotated TE in the genome together and counts a read if it aligns to any of them at least once, allowing proper quantitation of reads that would otherwise be discarded because of mapping to multiple locations. Read count tables from RepEnrich were processed with the edgeR package to perform normalization and calculate log$_2$ fold change. For small RNA-seq libraries, reads were aligned using RepEnrich and counts were normalized using unique alignments to cisNATs and structured loci, as described in ref. 11. Log$_2$ fold changes ($piwi^{-/-}$ / $piwi^{+/-}$) were calculated using normalized read counts. TEs with RNA-seq log$_2$ fold changes $>0.263$ (1.2× fold increase) in *piwi* mutants compared with heterozygous controls were plotted on the heat map together with the corresponding piRNA-seq change for each element. Heat maps were generated with the gplots package in R.

To create TE alignment profiles (Supplementary Fig. 4), reads were mapped uniquely to the relevant TE Repbase consensus sequence using Bowtie ($-v$ 1 $-m$ 1). Read depth across TE consensus sequence was determined using bedtools genomecov command, using total uniquely mapping reads in the library to normalize for library size differences. Plots were created in R.

For normal coding gene analysis, edgeR was used to determine differentially expressed genes whose expression was significantly changed in total RNA-seq libraries of *piwi* mutant fat body relative to heterozygous controls. We then performed a KEGG pathway analysis on these genes using Flymine.org[37].

**piRNA cluster analysis.** To determine abundance of small RNA reads mapping to annotated piRNA clusters, 23–29 bp size selected reads were aligned uniquely to the genome using Bowtie ($-v$ 1 $-m$ 1). piRNA cluster-specific reads were extracted using bedtools using the 15 most highly expressing piRNA clusters, as defined in refs 10,11. Specific cluster coverage plots were calculated using bedtools genomecov, normalizing to total uniquely aligning reads (excluding small nucleolar RNA, transfer RNA and miRNA), and plotted with the Sushi R package from Bioconductor[38].

To assay genic piRNA reads, 23–29 bp aligned reads were separated into 5′UTR, coding sequence and 3′UTR regions for each gene (Flybase annotations) and counted and sorted sense/antisense using bedtools. For selected genes with high numbers of antisense piRNAs in the 3′UTR (see, for example, Fig. 2g), coverage was calculated as described above. Coverage and gene models were plotted using Sushi[38].

**Immunoblotting.** Fly body segments and tissues were dissected as in 'RNA/smRNA-seq Library Preparations.' Whole-cell lysate samples were prepared in RIPA buffer (50 mM Tris-HCl, pH 7.5, 150 mM NaCl, 1 mM EDTA, 1 mM dithiothreitol, 1% Triton-X-100, 1% Na-deoxycholate, 0.1% SDS and 2.5 μg ml$^{-1}$ of Pepstatin A, Leupeptin, Antipain, Aproptinin and Chymostatin). Then, 15 μg of total protein was loaded and run on a 12% polyacrylamide–SDS gel. Proteins were then transferred to polyvinylidene difluoride membrane and the membrane was blocked overnight in Tris-buffered saline with 0.1% Tween-20 in 5% milk at 4 °C. Membranes were first cut between a 75 and 50 kDa marker and the lower molecular weight half of the membrane incubated with anti-Actin (mouse 1:2,000; EMD Millipore MAB1501). The upper molecular weight half of the membrane was incubated with anti-Piwi (mouse 1:200; Santa Cruz sc-390946), anti-Aubergine (rabbit 1:2,000) or anti-AGO3 (rabbit 1:3,000). Before blotting with either anti-Aubergine or anti-AGO3 and between each reprobing, the blot was stripped with stripping buffer (50 mM Tris-HCl, pH 6.8, 20 g l$^{-1}$ SDS and 0.8% β-mercaptoethanol) at 60 °C. After treating with each primary antibody, membranes were incubated with either horseradish peroxidase-conjugated anti-mouse (goat 1:5,000; ThermoFisher 31430) or anti-rabbit (goat 1:5,000; ThermoFisher 31460). The anti-AGO3 and anti-Aubergine antibodies were gifts from Phillip Zamore. Uncropped images of the original immunoblots can be found in Supplementary Fig. 2.

**Immunofluorescence assays and lipid staining.** Flies were briefly dipped in EtOH and blotted, fixed in 4% paraformaldehyde in PBS for 20 min, washed with PBS on ice and embedded in Tris-buffered saline Tissue Freezing Medium (Fisher Scientific 15-183-13). Moulds were frozen on dry ice and stored at $-80$ °C. Moulds were cryosectioned into 10 μm sections and gently washed with PBS. For anti-Piwi immunofluorescence, slides were incubated with anti-Piwi antibody (mouse 1:50; Santa Cruz sc-390946) for 1 h at room temperature, washed with PBS, incubated with anti-mouse Alexa Fluor-568-conjugated antibody (goat 1:2,000; ThermoFisher A-11031) for 1 h at room temperature and then washed again with PBS. Anti-Piwi primary and secondary antibodies were both diluted in 5% normal

goat serum and 0.1% Triton X-100. For lipid staining, sections were incubated with Nile Red (0.5 µg ml$^{-1}$; ThermoFisher N-1142) for 1 h at room temperature and washed with PBS. For anti-γ-H2A.v immunofluorescence, slides were washed in PBS and blocked for 30 min at room temperature in 5% normal goat serum and 0.5% Triton X-100. Slides were then briefly rinsed in PBS, incubated with anti-γ-H2A.v antibody (mouse 1:20; DSHB unc93-5.2.1) for 2 h at room temperature, washed with PBS + 0.1% Tween-20 (PBST), incubated with anti-mouse Alexa Fluor-568-conjugated antibody (goat 1:1,000; ThermoFisher A-11031) for 1 h at room temperature and washed twice with PBST followed by one final wash with PBS. Anti-γ-H2A.v primary and secondary antibodies were both diluted in 5% normal goat serum and 0.1% Tween-20. All slides were mounted and stained with 4,6-diamidino-2-phenylindole (DAPI; ThermoFisher P36935).

Images of Piwi were acquired with a Zeiss LSM 510 meta-confocal laser-scanning microscope and γ-H2A.v and lipid staining with a Zeiss AxioImager.Z1 ApoTome microscope. Images of representative nuclei, proteins and lipids were selected and processed using ImageJ and Adobe Photoshop. γ-H2A.v intensity was measured and quantified using ImageJ according to the method published in ref. 28. Lipid droplet size was measured and quantified using ImageJ from sections of individual flies. For γ-H2A.v fluorescence intensity and lipid droplet size quantification, Student's two-tailed t-test was performed to determine significance.

**gypsy-TRAP transposition reporter.** The piwi mutant fat body gypsy-TRAP reporter line was generated using the lines mentioned in 'Fly Stocks and Husbandry.' Briefly, the fat body-specific r4-GAL4 driver line was first recombined with a UAS-GFP line generating a new stable line that expresses green fluorescent protein (GFP) exclusively in the fat body. This line was then crossed with the gypsy-TRAP reporter line[24], a line containing a GAL80 driven by a tubulin promoter separated by an ovo binding site that attracts the gypsy TE that suppresses GFP expression until TE transposition activates reporter function. Once stable, this new line was then further crossed into a piwi[2]/CyO mutant background, thereby generating the final piwi mutant fat body gypsy-TRAP reporter line. Mixed sex heterozygous and homozygous flies from this line were then aged together for 10 days. Flies were then separated by piwi genotype and female flies of each piwi genotype further separated under a fluorescent dissecting scope into three groups according to the approximate number of GFP cells visibly fluorescing beneath the ventral abdominal wall. Groups were as follows: low (no visible GFP-positive cells), medium (<50% abdominal area showing GFP-positive cells) and high (>50% abdominal area showing GFP-positive cells). Fisher's exact test was performed to determine significance between the combined medium + high groups and the low group.

As was demonstrated for adult mushroom body neurons[24], we found that the GFP-positive cells in the fat body were dependent upon endogenous gypsy insertion as most ovo binding sites in the wild-type gypsy-TRAP/r4-GAL4::UAS-GFP reporter line resulted in no GFP-positive cells as expected[9]. All representative images were taken from heterozygous piwi mutant fat body gypsy-TRAP flies.

**Metabolic assays.** Assays were performed as in ref. 39 with the following modifications. Biological replicates for each genotype assayed were generated using five whole adult flies. All data were normalized to total protein concentration and calculated as a percent relative to the control genotype. For each assay, Student's two-tailed t-test was performed to determine significance.

**TAG assay.** Flies were homogenized in 200 µl of cold PBST buffer. The homogenate was heat-treated at 70 °C for 5 min to inactivate endogenous enzymes. Protein concentration was measured using a Bradford assay (Bio-Rad 5000006). Samples were diluted with PBST and 15 µl of heat-treated homogenate or glycogen standard (Standbio 2103-030) were incubated with 200 µl of Triglyceride Reagent (Thermo Scientific TR22421) for 5 min at 37 °C. Absorbance was measured at 540 nm and TAG content of samples was calculated based on a standard curve of TAG that was run in parallel with experimental samples.

**Glycogen assay.** Flies were homogenized in 200 µl of cold PBS. The homogenate was heat-treated at 70 °C for 5 min to inactivate endogenous enzymes. Samples were centrifuged for 3 min at 16,100 × g and the supernatant collected. Protein concentration was measured using a Bradford assay (Bio-Rad 5000006). Samples were diluted with PBS and 90 µl of heat-treated homogenate were incubated with either 20 µl of amyloglucosidase (Sigma-Aldrich A7420) or 20 µl of PBS. To create a glycogen standard curve, 50 µl of glycogen standards (Ambion AM9510) were treated with either 50 µl of amyloglucosidase or 50 µl of PBS. All samples were incubated at 37 °C for 1 h. Then, 30 µl of each sample was added to a 96-well plate. Next, 100 µl of Infinity Glucose Hexokinase reagent (Thermo TR15421) was added to all samples and incubated at room temperature for 15 min. The absorbance of samples was then measured at 340 nm and normalized by subtracting the absorbance of the free glucose of untreated samples from the absorbance of the total amount of glucose present in samples treated with amyloglucosidase. Glycogen content was then calculated based on the normalized glycogen standard curve.

**Trehalose assay.** Flies were homogenized in 200 µl of cold trehalose buffer (5 mM Tris-HCl, pH 6.6, 137 mM NaCl and 2.7 mM KCl). The homogenate was heat-treated at 70 °C for 5 min to inactivate endogenous enzymes. Samples were centrifuged for 3 min at 16,100 × g and the supernatant collected. Protein concentration was measured using a Bradford assay (Bio-Rad 5000006). Samples were diluted with trehalose buffer and 90 µl of heat-treated homogenate were incubated with either 20 µl of trehalase (Sigma-Aldrich T8778) or 20 µl of trehalose buffer. To create a trehalose (Sigma-Aldrich T9531) standard curve, 50 µl of standard were incubated with either 30 µl of trehalase or 30 µl of trehalose buffer. A free glucose (Fisher Scientific 50-99-7) standard curve was also generated. All samples were incubated at 37 °C overnight. Then, 30 µl of each sample was added to a 96-well plate. To all samples, 100 µl of Infinity Glucose Hexokinase reagent (Thermo TR15421) was added and incubated at room temperature for 15 min. The absorbance of samples was then measured at 340 nm and normalized by subtracting the absorbance of free glucose present in untreated samples from the total amount of glucose present in samples treated with trehalase. Trehalose and free glucose content were calculated based on standard curves of trehalose and glucose respectively.

**Glucose assay.** Flies were homogenized in 200 µl of cold PBS. The homogenate was heat-treated at 70 °C for 5 min to inactivate endogenous enzymes. Samples were centrifuged for 3 min at 16,100 × g and the supernatant collected. Protein concentration was measured using a Bradford assay (Bio-Rad 5000006). Samples were diluted with PBS and 30 µl of supernatant was then added to a 96-well plate. To all samples, 100 µl of Infinity Glucose Hexokinase reagent (Fisher Scientific TR15421) was added and incubated at room temperature for 15 min. The absorbance of samples was then measured at 340 nm and glucose content was calculated based on a standard curve of glucose (Fisher Scientific 50-99-7).

**Starvation assay.** Before permanent starvation, flies were synchronized in their feeding by fasting on 2% agar for 4 h followed by a final period of feeding for 2 h. Flies were then sorted under $CO_2$ anaesthesia into separate sex vials containing 2% agar at a density of 10 males or 10 females per vial, with a total of 5 vials ($n \approx 50$) for each genotype. Dead flies were scored and counted every 6 h. Starvation analyses were performed and log rank statistics calculated using the online application OASIS[40]. Starvation assays were repeated at least twice.

**Immune challenge assay.** A bacterial culture of Erwinia carotovora (Gram-negative insect/plant pathogen), a gift from Neal Silverman, was grown overnight in Luria broth, shaking at 220 r.p.m. on a platform shaker at 37 °C. The culture was allowed to reach $OD_{600nm} \approx 2.0$ before removing 1 ml for centrifugation at 2,000 × g for 2 min. The supernatant was removed and the bacterial pellet gently washed with 1 ml of 10 mM $MgSO_4$ to remove traces of culture media. The wash was removed and the pellet resuspended in 10 µl of fresh 10 mM $MgSO_4$. A 0.15 mm needle was then dipped into 80% EtOH and flame sterilized. Flies were sorted under $CO_2$ anaesthesia and inoculated by dipping the needle into either the bacterial suspension or EtOH for mock infection controls, and inserting the needle midway into one side of the thorax. Flies were then sorted into separate sex vials at a density of 10 males or 10 females per vial, with a total of 5 vials ($n \approx 50$) for each condition/genotype, and passed to fresh food at least every other day. Dead flies were scored and counted every 24 h. Immune challenge analyses were performed and log rank statistics calculated using the online application OASIS[40]. Immune challenge assays were repeated at least twice.

**Lifespan assay.** Flies used in lifespan experiments were collected upon eclosion over a 48 h period and were sorted under $CO_2$ anaesthesia and placed in food vials at a density of 25 males and 25 females per vial, with a total of 10 vials ($n \approx 250$) for each genotype. Lifespan food consists of 50 g l$^{-1}$ autolysed yeast, 50 g l$^{-1}$ sucrose and 20 g l$^{-1}$ agar (all w/v). For the flamenco 3TC lifespans, flamenco homozygous mutant flies were collected and aged on either food containing 150 g l$^{-1}$ autolysed yeast, 150 g l$^{-1}$ sucrose and 20 g l$^{-1}$ agar, (all w/v) or identical food with 10 mM lamivudine (3TC). Flies were passed to fresh food every other day, and dead flies scored and counted. Lifespan analyses were performed and Wilcoxon rank sum test statistics calculated using the online application OASIS[40]. Lifespan assays were repeated twice.

**Data availability.** The data sets generated and analysed in this study have been deposited and are available in the GEO database, under the series accession number GSE89260. Additional relevant data sets and computer code are available either in this published article (and its Supplementary Information Files) or are available from the corresponding author on reasonable request.

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

## Acknowledgements

We thank Gillian Horwitz, Mark Henriksen, Davis Hartnett, Jackson Taylor and Suzanne Hosier for technical assistance and Will Lightfoot for fly food preparation. We also thank Haifan Lin, Joshua Dubnau, Robert Reenan, Marc Tatar and the Bloomington *Drosophila* Stock Center for fly stocks; John Sedivy and Neal Silverman for reagents; Chengjian Li, Bo Han and Phillip Zamore for reagents and advice on bioinformatic analysis; Marissa Holmbeck, Leila Rieder, Yiannis Savva, Erica Larschan, Nicola Neretti and Robert Reenan for discussions and comments on the manuscript. This work was supported by a NIA T32 Training Grant AG041688 and a F31 Predoctoral Research Fellowship Award AG047736 to B.C.J. and NIA Grants AG16667 and AG24353, a Glenn/American Federation for Aging Research (AFAR) Breakthroughs in Gerontology award, NIH Program Project Grant AG51449 and Grant P30AI042853 from the Providence/Boston Center for AIDS Research to S.L.H.

## Author contributions

B.C.J. conceived and directed the project with contributions from J.G.W. and S.L.H. C.C. isolated all body segment and pure fat body samples. J.G.W. and C.C. performed all sequencing library preparations. J.G.W. performed all bioinformatic analyses with support from B.C.J. B.C.J. performed all immunoblotting. B.C.J. generated and assayed the piwi mutant fat body gypsy-TRAP flies. M.J.F., E.R.S. and B.C.J. performed all sectioning, immunofluorescence, lipid staining and microscopy. B.C.J. quantified immunofluorescence and lipid droplet staining. B.C.J. processed all immunoblot and microscopy images for publication. A.D.T. and B.C.J. performed all metabolic assays. M.J.F., E.R.S., A.D.T. and B.C.J. performed all starvation assays. E.R.S. and B.C.J. performed all immune challenge assays. C.C., E.R.S. and B.C.J. performed all lifespan assays. B.C.J. wrote the manuscript with assistance from J.G.W. and S.L.H.

## Additional information

**Competing financial interests:** The authors declare no competing financial interests.

