## [Peer Review File · Nature Communications]

Reviewers' Comments:

Reviewer #1 (Remarks to the Author)

Although the Piwi pathway is most highly expressed in the animal gonad and is thought to primarily repress TEs during gametogenesis, the role of this pathway in the soma remains less studied because expression is much lower outside of the gonad and the sterility is the most overt phenotype in Piwi pathway mutants. This manuscript represents one of the few newer studies to probe more carefully additional phenotypes beyond the gonad for the Piwi pathway, by investigating potential PIWI expression and piRNAs in the drosophila adult fat body. This study shows the expression of piwi pathway transcripts and protein from dissected fat body from eviscerated abdomen, and sequencing of piRNAs and total RNAs from the fat body of wt, heterozygous and homozygous piwi mutants. There are clearly signatures of piRNAs and there is the expected up-regulation of transposon transcripts in the homozygous mutant. This study then reports interesting fat and glycogen storage differences in the piwi mutant as well as decreased longevity and fitness to immune challenge and starvation in the homozygous mutant compared to the heterozygous control.

Overall, I find this study nicely presented and highly interesting to a readership at Nature Comm, however, I have two main concerns that I think are vital for buttressing the thesis of this study, important but missing controls to bolster that the results are truly reflecting a piwi pathway specifically acting in the Drosophila fat body. If the authors can address these two main issues, then I would be fully supportive of the acceptance for this study for publication.

First, the topic of significant piwi pathway expression in the soma still remains highly controversial, and I recognize the data presented in this study has gone to considerable lengths to argue for specific piwi protein and piRNA expression in the fat body. However, the authors have not controlled for the possibility that even a minute contamination of ovarian tissue into the dissected fat bodies, which in the abdomen are quite close to the ovaries, could also account for their results. Because the PIWI protein and piRNAs are so abundantly expressed in the ovary, the piRNA sequencing, the western blot of Fig1C and the expression of cytochrome and bicaudal in Extended Fig 1C could all be explained by a small amount of ovary contaminating the eviscerated abdomens. Unfortunately, using the piwi^{-/-} mutant in Fig 2 cannot resolve this issue because the mutation should theoretically affect piwi and piRNA expression from both the ovarian and fat body tissues.

However, a genetic experiment that could resolve this is to examine mutants that specifically ablate either the piwi pathway or kill development altogether just in the ovaries while leaving the rest of pathways intact in the adult fly abdominal fat body. This could be achieved with either an OvoD1 homozygous mutant or using Nos-Gal4 strongly driving RNAi against Piwi pathways. By using genetics to block Piwi completely in the ovary, then doing the western blot and small RNA sequencing from the whole fly, will piRNAs and Piwi protein still be detected? The data would also have to show extremely penetrant loss of ovaries in females of these mutants, and if PIWI and piRNAs persist, this effort could settle the controversy once and for all that Piwi is truly active outside of the gonad.

Secondly, another missing control for genetic rigor is that the current data in Figure 3 only use a single piwi2 allele for homozygous mutants. But as shown in by Jin et al, Curr Bio, 2013 there are additional alleles to generate trans-hets mutants to control for background effects that might be specific to just the piwi2^{-/-}. Did the authors examine piwi1 or piwi(delta37) as trans-hets with piwi2 in the aging and glycogen and lipid analyses? If these mutants are not available, there are also new piwi crispr mutants from Senti et al, Genes Dev 2015.

Having an additional piwi mutant allele is more critical for the glycogen and lipid-content assays in

Fig. 3, but I do note that this issue is less critical for Fig. 4, where in these aging curves in the piwi mutants could manifest decreased longevity fitness in the flies whether in the ovary or in the fat body. Also, the flam mutant provides a very nice reproducibility and background control since it is a different mutation affecting just the piRNAs that Piwi binds to, and thus in the same pathway, but should have a completely different genetic background from the piwi2 mutant. I think it is interesting and logical to see reduced longevity in the flam homozygous mutant, but this effect is still unclear if this is fat body specific or mediated through the ovary. Therefore, I would suggest to the authors to temper the text and figure legend of Fig4 not to state these effects are just fat body stressor alone, but could also be tied to oogenesis that may signal to the rest of the organism including the fat body during the aging assays.

Reviewer #2 (Remarks to the Author)

In light of the growing body of evidence showing mobility of TEs in the soma, studies into piRNA production and function in diverse somatic cell are timely and important. The present manuscript describes such a study in *Drosophila* fat body. The manuscript is concisely written and easy to follow. It reports two main findings: (i) that a functional piRNA pathway exists in *Drosophila* fat body, and (ii) this has physiological significance for lipid storage, stress tolerance and lifespan in flies.

This nicely comprehensive study spans investigations from RNA sequencing, protein localization, and TE activity to physiological impacts such as lipid storage levels, immunity and life span.

The results appear mostly convincing, but this reviewer has the following concerns:

1. Statistical analysis has been incompletely performed throughout the manuscript. This makes it difficult to judge the conclusions. For example, the number of replicates and p-values are not reported on Figures 1 and 2. It is not clear if the data shown in Fig 1a has been normalized for the amount of input material (e.g. cell mass or similar)? How was the heat map in Fig 2a calculated (e.g. based reads with the greatest increase or all reads)? Is the increase in TE levels significant? The associated extended data figure suggests that for most cases the increase is very small. Moreover, it would be good to show how non-TE transcripts behave in piwi $-/-$. Statistics are also missing for the other panels in Fig 2.
2. It is surprising that different essential piRNA pathway components are enriched at different levels in the eviscerated abdomen. In fact, some show only very small increase relative to the controls. How do the authors explain this?
3. Based on the presented data, this reviewer is not convinced that transposon activation is directly linked to the observed phenotypic effects. While it appears that piRNAs go down and transposons go up in piwi $-/-$ fat body, functional data on transposon mobilization is only shown for one isolated element. Also, the data on increased transposon expression is not very convincing - the levels of increase appear generally rather low-, while the phenotypic effects look rather strong. Could TE expression be a "side effect", but not causal?
4. Related to the previous, it is not clear through what mechanism TE activation would influence lipid storage levels and lifespan. While H2A.v staining implicates the occurrence of DNA breaks, replicative transposition does not normally induce high levels of dsDNA breaks (and the levels of transposon expression shown are not very high to start with, as mentioned above). To further confirm the role of DNA breaks and transposition in reduced fat body functionality, apoptosis of the relevant cells could perhaps be directly tested. Alternatively, the authors should rephrase their conclusions more carefully.
5. Lastly, do the authors have a hypothesis why fat body would need extra protection against TEs, while most other somatic cells are fine without it?

We appreciate the constructive comments provided to us by the reviewers and are thankful for the opportunity to submit our revised manuscript. The reviewers showed enthusiasm for our manuscript while also asking for experiments with additional genetic controls, specific statistical analyses, and modifications to the text. We have taken these comments into consideration, performed additional experiments, and have clarified the text. These changes have significantly improved our manuscript and we address each of the reviewers' points below.

Reviewer #1

1. The reviewer raises the concern that observation of Piwi protein and piRNAs in the adult fat body could be a result of contamination from ovarian tissues and suggests the use of a genetic mutant lacking ovaries, *ovo^{D1}*, to control for this potential issue. We have since sequenced oxidized fat body libraries generated from *ovo^{D1}* animals and identified the presence of fat body piRNAs that exhibit characteristics similar to those found in wild type fat bodies (Extended Data Figure 5). We believe that these new data, combined with our observation of Piwi protein specifically in the nuclei of fat body cells using fluorescent immunohistochemistry, strongly support our observation of a piRNA pathway in the fat body that is not due to ovarian contamination.

Text: lines 93-104

In order to rule out the possibility that fat body piRNAs result from contamination by ovarian tissues during dissection and library preparation, we performed smRNA-seq on oxidized fat body libraries isolated from *ovo^{D1}* flies. Due to a dominant female-sterile mutation in the *ovo* gene, these flies exhibit severely degenerated ovaries thus significantly decreasing the likelihood that any piRNAs isolated from the fat body of these animals would be due to contamination from ovarian tissues. We observed in both wild type and *ovo^{D1}* fat bodies 23-29nt smRNAs that mapped uniquely to the *flamenco* locus (Extended Data Fig. 5a, b), suggesting that piRNAs observed in oxidized fat body libraries are in fact originating from this tissue and not a result of ovarian contamination. Additionally, immunoblotting demonstrates the presence of Piwi protein in the eviscerated abdomen and isolated fat body of *ovo^{D1}* flies (Extended Data Fig. 5c). These data, combined with our observation of the Piwi protein in the nuclei of fat body cells (Fig. 1b), strongly support the presence of a functional canonical piRNA pathway in the fat body.

2. The reviewer requested that we repeat TAG and glycogen metabolite assays using an additional piRNA pathway mutant to account for possible effects contributed by different genetic backgrounds of individual fly lines. We have performed both the TAG and glycogen assays using *flamenco* mutants and have observed similar significant decreases that we observed in *piwi* mutants (Fig. 3f-i). We chose *flamenco* mutants, as they are a completely different genetic background from *piwi*, the mutation is specific to TE reactivation (i.e. loss of piRNA cluster function), and they will complement our lifespan assays using this mutant (Fig. 4d). It is therefore unlikely that these metabolic phenotypes are due to a genetic background effect.

Text: lines 120-122

These data correlate with a significant reduction of two of the major storage metabolites in fat body, triacylglycerides (TAGs) and glycogen, in both *piwi* and *flamenco* mutants (Fig. 3f-i and Extended Data Fig. 6a, b).

The reviewer also notes the possibility that our observation of a shortened lifespan in piRNA pathway mutants could be due to an effect on oogenesis and therefore suggests a tempering of the text and figure legend of Figure 4. We have made these changes and also emphasize in the text that a lifespan extension and accumulation of lipid stores is often observed in animals with compromised or ablated ovarian function, which is opposite to the phenotypes we have observed, therefore suggesting compromised fat body function unrelated to ovarian dysfunction.

Text: lines 152-159

While we are not able to entirely rule out a contribution of the gonadal piRNA pathway to fat body function, many of the phenotypes we observe are opposite to those typically seen in animals with compromised gonadal tissue function and therefore likely represent the effect of a loss of the fat body piRNA pathway. For example, the shortened lifespan and reduced lipid stores in piRNA pathway mutants demonstrates that the piRNA pathway is essential in the health and functioning of non-gonadal somatic tissues, since reduction or ablation of gonadal function in flies often extends lifespan and increases lipid stores rather than decreasing lifespan and fat storage²³.

Reviewer #2

1. The reviewer raises questions about the statistical approaches for various figures. The reviewer requests replicate numbers and relevant *P*-values. In Figure 1, for each RNA-seq experiment, three independent biological replicates were used as described in the Methods section. Pairwise FDR values, calculated using edgeR, are now reported for all genes and comparisons in Fig. 1a and Extended Data Figure 1 in a supplementary table (Extended Data Table 1) and described in the corresponding figure legends. The reviewer also requested that we make clear the method of normalization for Fig. 1a. As RNA-seq data cannot be properly normalized based on input material, we have reported the relative expression levels using the standard RPKM (reads per kilobase per million) metric, which corrects for differences in sequencing library size among samples and replicates. Regarding the reviewer's question for the methodology of Fig. 2a, we have further described this methodology for the heat map in the Methods section. The reviewer also asks whether changes in TE levels are significant in Fig. 2a. The heat map shows the log₂ fold change between *piwi*^{-/-} and *piwi*^{+/-} flies for TEs >1.2 fold change, and is ordered by largest TE fold change. We have also now labeled TE fold changes that are significant at a threshold of FDR<0.05 (calculated by edgeR) with an asterisk and reflected this in the figure legend. These changes are also now detailed in the Methods section. The reviewer asks that we show how non-TE transcripts behave in *piwi* mutants. We have now included an extended data figure showing a KEGG term analysis (Extended Data Figure 6c) based on our total RNA-seq dataset of significant differentially expressed coding genes. The reviewer notes that statistical analyses are also missing for other panels in Fig. 2. For Fig. 2c, we now report a *P*-value using a Fisher's exact test showing the two genotypes are significantly different. For the panels showing small RNA sequencing data, (Fig. 2b, d, and f), we have combined together our sequencing libraries and reported the normalized data together. Because of the relatively smaller yield and lower read depth of small RNA-seq libraries compared to total RNA libraries, it is not possible to accurately quantify statistical variance from these smRNA libraries. However, our methodology and data reporting for smRNAs are consistent with other publications from leaders in the field, including [Czech et al. (*Molecular Cell*) 2013, Handler et al. (*Molecular Cell*) 2013, Han et al. (*Bioinformatics*)

2014, Han et al. (*Science*) 2015, Yu et al. (*Science*) 2015, Iwasaki et al. (*Molecular Cell*) 2016.]

2. The reviewer asks why different piRNA pathway components are enriched at different levels in the eviscerated abdomen and notes that many show a relatively small increase relative to somatic controls. To address this, we have added an extended data table (Extended Data Table 1) with the FDR values of all of the genes displayed in Fig. 1a and Extended Data Fig. 1. In comparing the abdomen data with the head and thorax controls, the increases we observe are statistically significant in 10 of 11 genes displayed in Fig. 1a, the exception being *tj*, which codes for a transcription factor that is important for *piwi* transcription, but also has other genomic targets (see Extended Data Table 1). Likewise, in Extended Data Fig. 1a, of the 14 genes presented, 11 are statistically significantly increased in abdomen vs. head/thorax, one is significantly lower in abdomen vs. thorax (*Ci*), and two are not significantly different (*fs(1)Yb* and *Tudor*) (see Extended Data Table 1). Because of tissue and cell type heterogeneity as well as the transcriptomic demands of the cell, we do not necessarily expect all genes in this pathway to be upregulated or expressed in an identical manner. Finally, different piRNA pathway genes are expressed at different levels in ovarian tissues, as we have observed in our ovarian control libraries (Fig. 1a and Extended Data Fig. 1), and are required at different stoichiometric ratios, so we are not surprised that this is also the case in the fat body.

3. The reviewer notes that we have shown mobilization of one TE, the *gypsy* element, and have not causally shown that TE reactivation results in compromised fat body function. We attempt to address this concern by including additional lifespan assays where we have employed the reverse transcriptase inhibitor 3TC to prevent TE retrotransposition. These assays show a lifespan extension of *flamenco* mutants fed 3TC suggesting that their shortened lifespan may be partially due to TE activity. We also emphasize that *gypsy* is a known target of the piRNA pathway and an excellent indicator of its activity [Handler et al. (*Molecular Cell*) 2013], therefore, our use of the *gypsy-TRAP* reporter is particularly relevant. We have also tempered the language in the text of the manuscript to deemphasize causation and instead highlight a correlation between TE activity and loss of fat body function.

Text: lines 136-146

Finally, we tested whether the shortened lifespan of piRNA pathway mutants was dependent upon TE activity. Many TEs in *Drosophila* are retrotransposons, including *gypsy*, and depend upon reverse transcriptase for their replicative mobility²⁴. Administration of a known reverse transcriptase inhibitor, 3TC, inhibits the normal age-related increase in *gypsy* mobilization and extends the shortened lifespan of *Dcr-2* mutants, another condition in which de-repression of TEs occurs¹⁶. We administered 3TC to *flamenco* mutants and observed a significant lifespan extension (Fig. 4e and Extended Data Table 2) suggesting that a shortened lifespan phenotype is at least partially dependent upon TE mobilization. These results suggest that loss of the fat body piRNA pathway and an increase in TE activity and mobilization correlates with compromised fat body function including its ability to otherwise mitigate the detrimental effects of environmental stressors.

4. The reviewer expresses concern regarding the mechanism by which TE activation would influence lipid storage and lifespan and specifically references our observation of elevated levels of gamma-H2A.v staining. We note that other groups, ours included,

have shown in both flies and human cell culture that reactivation of TEs in models of elevated TE activity correlates with increased levels of dsDNA breaks [Wood et al. (*PNAS*) 2016, Chen et al. (*Aging Cell*) 2016, Gasior et al. (*J. Mol. Biol.*) 2014]. Per the suggestion of the reviewer, we have also rephrased our conclusions to deemphasize causation.

Text: lines 108-113

Phosphorylation of the histone variant H2A.v (γ -H2A.v) during DNA repair serves as a reliable marker of DSBs and has been shown to correlate with increased TE activity in the fat body^{16,20}. Using immunofluorescent microscopy, we observed an increase in the intensity of γ -H2A.v staining in *piwi* mutants relative to controls (Fig 3a, b). These data suggest that the fat body piRNA pathway normally protects fat body cells from the accumulation of DNA damage that may be caused by TE reactivation.

5. The reviewer asks that we provide a hypothesis as to why the fat body would require a piRNA pathway. Investigation of this topic is presently outside the scope of this report, but we put forth a few hypotheses that may explain the existence of the fat body piRNA pathway. Multiple TEs in flies are thought to be retroviral-like [Touret et al. (*Viruses*) 2014, Lecher et al. (*J. Gen. Virol.*) 1997, Kim et al. (*PNAS*) 1994], suggesting they could be secreted into the hemolymph from the fat body and taken up by the developing ovarian cells. In this manner, the fat body piRNA pathway, similar to the piRNA pathway in the somatic ovarian follicle cells, may serve as an important additional defense against vertical transmission of TEs. It is also interesting to note that the fat body and somatic follicle cells of the ovary share a common developmental lineage [Riechmann et al. (*Development*) 1998, Moore et al. (*Development*) 1998], which may help explain the presence of a piRNA pathway in both of these tissues.

Reviewer #1 (Remarks to the Author)

I have now thoroughly reviewed the revised manuscript and author rebuttal. I am satisfied with the changes and support the publication of this manuscript.

Reviewer #2 (Remarks to the Author)

The authors have gone a long way to address all reviewer comments and have certainly answered all my concerns and comments satisfactorily. I support publication of this nice work.